# The Challenge to the Pathologist of PD-L1 Expression in Tumor Cells of Non-Small-Cell Lung Cancer—An Overview

**Korinna Jöhrens** [1,*] **and Josef Rüschoff** [2]

1    Institute of Pathology, Carl Gustav Carus University Hospital Dresden, 01307 Dresden, Germany
2    Targos Molecular Pathology GmbH, 34119 Kassel, Germany; rueschoff@patho-nordhessen.de
*    Correspondence: korinna.joehrens@uniklinikum-dresden.de; Tel.: +49-0-351-458-3041

**Abstract:** In recent years, the treatment of non-small-cell lung cancer (NSCLC) has been fundamentally changed by immunotherapy where the immune system is reactivated using anti-programmed cell death protein 1/programmed death ligand 1 (PD1/PD-L1) checkpoint inhibition. With this, the immunohistological detection of PD-L1 has become one of the most important predictive biomarkers, leading pathologists to play a central role in the immuno-oncological therapy decisions. This has brought them the challenge of requiring the knowledge of relevant checkpoint inhibitors (CI), different PD-L1 scores and cut-offs as well as the choice of the right tissues and controls. Their involvement is also required in the careful validation of both clinical trial assays (CTAs) and laboratory developed tests (LDTs), in addition to the internal and external quality assessment and the interpretation and scoring of the staining based on specialist training. After the training of tumor proportion score (TPS) scoring in NSCLC, pathologists show a high level of concordance, with some variation between different cut-offs. Since not all patients benefit from immunotherapy, further research is needed to validate new predictive markers and optimize existing ones. In this context, these studies focus on a combination of PD-L1 and molecular signatures.

**Keywords:** PD-L1; CTA; LDT; preanalytics

## 1. Introduction

In recent years, two major therapeutic advances have been achieved in non-small-cell lung cancer (squamous and nonsquamous NSCLC). One relates to personalized medicine and is essentially based on molecular alterations such as mutations in the EGFR, KRAS or BRAF genes, rearrangements of EML-ALK, amplifications in the MET gene, fusions in ROS1 and mutations in the PI3KCA-mTOR pathway [1]. The other concerns immunotherapy where the immune system is reactivated through checkpoint inhibition. In this therapeutic approach, the specific antibodies used act via immune checkpoint blockade.

In the case of the PD-1/PD-L1 pathway, the ligand–receptor pairs are located either on the immune cells or on the tumor cells [2]. The programmed cell death protein 1 (PD-1), programmed death ligand 1 PD-L1 and programmed death ligand 2 PD-L2 belong to the B7/CD28/CTLA-4 receptor family, which have a membranous, transmembrane and cytoplasmic component [3].

The cytotoxic T-lymphocyte-associated antigen (CTLA-4) and PD-1 are negative regulators of the T-cell immune function [3]. Therefore, for example, the binding between PD-1 and PD-L1 leads to an inhibitory effect that affects the T-cell response. In the healthy body, this process is important to maintain in the context of immune tolerance and the prevention of autoimmunity. However, as far as tumor cells or tumor infiltrating cells are concerned, this inhibition mechanism may be utilized by the tumor to suppress the immune response against itself, leading to its immune escape [4]. This immunosuppressive situation allows the tumor to grow uncontrollably. By applying checkpoint inhibitors, this mechanism can in turn be interrupted or turned off, so that the immune response against the tumor is re-established and strengthened [5].

The checkpoint molecules CTLA4 and PD-1 can lead to an exhaustion of T-cells, thereby causing the T-cell function to decrease with an increased expression of immune checkpoints [6,7].

Whereas CTLA4 interacts with its ligands during the early T-cell priming stage, PD-1 and PD-L1 suppress activated T-cells at the effector phase [8]. Moreover, CTLA-4 is confined to T-cells whereas PD-1 is more broadly expressed not only in activated T-cells, but also in B-cells and myeloid cells [3].

The intracytoplasmic component includes an immunoreceptor tyrosine-based inhibitory motif (ITIM) as well as an immunoreceptor tyrosine-based switch motif (ITSM). The ITSM seems to be essential for PD-1 function in T-cells and B-cells. Studies investigating mutations in this area have shown that the inhibitory function appears to be dependent on the function of ITSM phosphotyrosine, which leads to the downregulation of downstream pathways [9].

Regarding NSCLC, three different inhibitors are used therapeutically: anti-CTLA4, anti-PD-1 and anti-PD-L1 antibodies, which attack at different points in the immune system. In this context, it is important to note that the administration of CI has therapeutic relevance only in NSCLC and not in SCLC [10].

## 2. Approvals

### 2.1. Relevant Checkpoint Inhibitors

Several clinical studies have demonstrated that PD-L1 expression by tumor cells and immune cells is of predictive value concerning the efficacy of immunotherapy using both PD-1 and PD-L1 inhibitory drugs. It is therefore used as a predictive biomarker [11] such that for each of the different inhibitors, a specific immunohistochemical assay combined with different scores and cut-off has been developed as a so-called clinical trial assay (CTA). This poses a challenge not only for clinicians but also for pathologists, which will be highlighted below.

The following immune checkpoint therapies are relevant to NSCLC in stage III and IV as a first- or second-line treatment, as monotherapy or in combination with other IC or chemotherapy (see Table 1).

**Table 1.** Immunotherapy relevant to NSCLC.

| Drug | Therapy-Line | PD-L1 Test (CTA) | Score | Cut Off | Therapy Type |
|---|---|---|---|---|---|
| **Atezolizumab** | 1L stage IV<br>2L stage IV | Yes (SP142)<br>None None | TC or IC | ≥50% TC or ≥10% IC | Monotherapy<br>Monotherapy |
| **Cemiplimab** | 1L stage III<br>1L stage IV | Yes (22C3) | TPS | ≥50% | Monotherapy |
| **Durvalumab** | 2L stage III | Yes (SP263) | TPS | ≥1% | Monotherapy |
| **Ipilimumab** | 1L stage IV | None | | | in combination with Nivolumab and 2 cycles of platinum-based chemotherapy |
| **Nivolumab** | 1L stage IV | None | | | in combination with Ipilimumab and 2 cycles of platinum-based chemotherapy |
| | 2L stage IV | None | | | Monotherapy |
| **Pembrolizumab** | 1L stage IV | Yes (22C3) | TPS | ≥50% | Monotherapy |
| | 1L stage IV | None | | | in combination with pemetrexed and platinum-based chemo- therapy in non squamous NSCLC or in combination with carboplatin and paclitaxel or nab-paclitaxel in squamous NSCLC |
| | 2L stage IV | Yes (22C3) | TPS | ≥1% | Monotherapy |

Legend: CTA: clinical trial assay.

Cemiplimab is an anti-PD-1 antibody that has been approved as monotherapy for the first-line treatment of adult patients with NSCLC expressing PD-L1 and without EGFR, ALK or ROS1 aberrations. This drug is intended for patients who have locally advanced NSCLC but are not suitable for definitive radio-chemotherapy or for patients with metastatic NSCLC. The approval study was performed using the TPS with a cut-off of ≥50% and was assessed by an IHC 22C3 pharmDx assay.

Durvalumab is an anti-PD-L1 antibody that was approved with a VENTANA PD-L1 (SP 263) assay and is indicated as monotherapy for the treatment of locally advanced, unresectable non-small-cell lung cancer (NSCLC) in patients who have PD-L1-positive NSCLC that has not progressed following platinum-based radio-chemotherapy. As with Cemiplimab, only tumor cells, and not immune cells, are evaluated for PD-L1 expression, so the TPS applies here as well. The cut-off is ≥1% of tumor cells.

Pembrolizumab is an anti-PD-1 antibody. This immune checkpoint inhibitor is indicated as monotherapy for the first-line treatment of PD-L1-positive NSCLC without EGFR or ALK-positive tumor mutations. The positivity is defined as follows: a tumor proportion score (TPS) of ≥50% and assessed by a 22C3 pharmDx assay. Moreover, Pembrolizumab is approved as a second-line therapy for patients with NSCLC and PD-L1 positivity, where the cut-off is ≥1% scoring of the tumor cells. Finally, Pembrolizumab can be given, regardless of PD-L1 status, in combination with pemetrexed and platinum-based chemotherapy in nonsquamous NSCLC or in combination with carboplatin and paclitaxel or nab-paclitaxel in squamous NSCLC.

Atezolizumab is an anti-PD-L1 directed drug and has been approved as first- and second-line therapy for lung cancer regardless of PD-L1 status (determined by VENTANA PD-L1 (SP 142 assay). In addition, it was recently approved as a first-line treatment for adults with PD-L1-positive metastatic non-small-cell lung cancer (NSCLC) without EGFR (epidermal growth factor receptor) mutations or with ALK (anaplastic lymphoma kinase)-negative NSCLC. Here, the positivity is defined as PD-L1 expression in ≥50% of tumor cells (TC) or ≥10% in tumor-infiltrating immune cells (IC). This is currently the only CI for which the expression of PD-L1 on the immune cells also plays a role in the application.

Ipilimumab is an anti-CTLA4 antibody and has been approved independent of PD-L1 status (assessed by a 28-8 pharmDx assay) for the first-line treatment of metastatic NSCLC in patients who do not have a sensitizing EGFR mutation or ALK translocation in combination with nivolumab and 2 cycles of platinum-based chemotherapy.

Nivolumab is an anti-PD-1 antibody approved with PD-L1 IHC 28-8 pharmDx (independent of PD-L1 status), which is indicated as a first-line treatment for metastatic non-small-cell lung cancer (NSCLC) in patients whose tumors do not have a sensitizing EGFR mutation or ALK translocation, given in combination with ipilimumab and 2 cycles of platinum-based chemotherapy. Moreover, it is also indicated as monotherapy for the treatment of locally advanced or metastatic non-small-cell lung cancer after prior chemotherapy in adult patients.

### 2.2. FDA and EMA

For the case where PD-L1 expression is predictive of a response to a checkpoint inhibitor, FDA combines the approval of the drug with the immunohistochemical assay that has specifically been used in the clinical trial (companion diagnostic). In contrast, EMA allows the application of checkpoint inhibitors without specifying the companion diagnostic assay. This has led to the application of immunohistochemical staining protocols developed by pathology laboratories (LDT), which form the basis for PD-L1 expression analysis and are thus critical to the application of immunotherapy.

### 3. Relevant PD-L1 Scores

There are currently two PD-L1 scoring methods used in NSCLC. On the one hand, there is the TPS, and on the other hand, there are the TC and IC. In contrast to other carci-

noma entities such as head and neck squamous cell carcinoma (HNSCC), the determination of the combined positive score (CPS) has no therapeutic relevance in NSCLC.

The definitions of the two different scores are as follows:

### 3.1. TPS/TC

Depending on the immune checkpoint inhibitor to be administered, PD-L1 expressing tumor cells are reported in terms of their TC or TPS status. Nevertheless, in practical terms, both scores are to be considered identical.

In the tumor proportion score (TPS) or tumor cell (TC) score, PD-L1-positive tumor cells are evaluated in relation to all viable tumor cells on the slide. It is therefore a score based on tumor cell numbers. At least 100 tumor cells are required for the evaluation. The TC is a percentage value that can be additionally subdivided into four cut-off groups:

TC 0: 0~<1%;
TC 1: ≥1%~<5%;
TC 2: ≥5%~<50%;
TC 3: ≥50%.

### 3.2. IC

The Immune Cell Score (IC) indicates the proportion of the tumor that is occupied by PD-L1-positive tumor-infiltrating immune cells (lymphocytes, macrophages, granulocytes, dendritic cells, plasma cells).

The percentage is scored as follows:

IC 0: 0~<1%;
IC 1: ≥1%~<5%;
IC 2: 5%~<10%;
IC 3: ≥10%.

As mentioned above, a positive TC or IC Score is defined as the PD-L1 expression in more than 50% of tumor cells (TC) or more than 10% in tumor-infiltrating immune cells (IC), which translates to a score of TC 3 or IC 3.

## 4. General Aspects in the Interpretation, Training, and Reporting of PD-L1 Staining

### 4.1. Interpretation of PD-L1 Staining

Regardless of the scoring method used, there are several points to consider in the immunohistochemical evaluation of PD-L1.

Regarding the evaluation of tumor cell staining, it must be taken into account that not only the intensity but also the pattern of subcellular expression itself can be quite different.

In general, a carcinoma cell is considered as PD-L1-positive if the cell membrane is partially or completely stained, irrespective of the staining intensity [12]. Therefore, the tumor cells can have a basolateral membrane staining that is incomplete (Figure 1A–D). The cytoplasmic staining of tumor cells alone is not sufficient for a positive interpretation (Figure 2).

In contrast, any staining, membranous or cytoplasmic, is accepted as a positive score in immune cells (Figure 3). In situ necrosis and carcinoma as well as alveolar macrophages must, however, be excluded from evaluation (Figure 4). It is worth noting that the latter could be helpful as an internal control, while a nuclear expression can be an indication of a fixation problem (Figure 5). Moreover, the staining of stroma elements or basement membrane should be disregarded. While endothelia should be negative in the positive control, endothelia surrounding the tumor may occasionally be positive. These should not be included in the scoring.

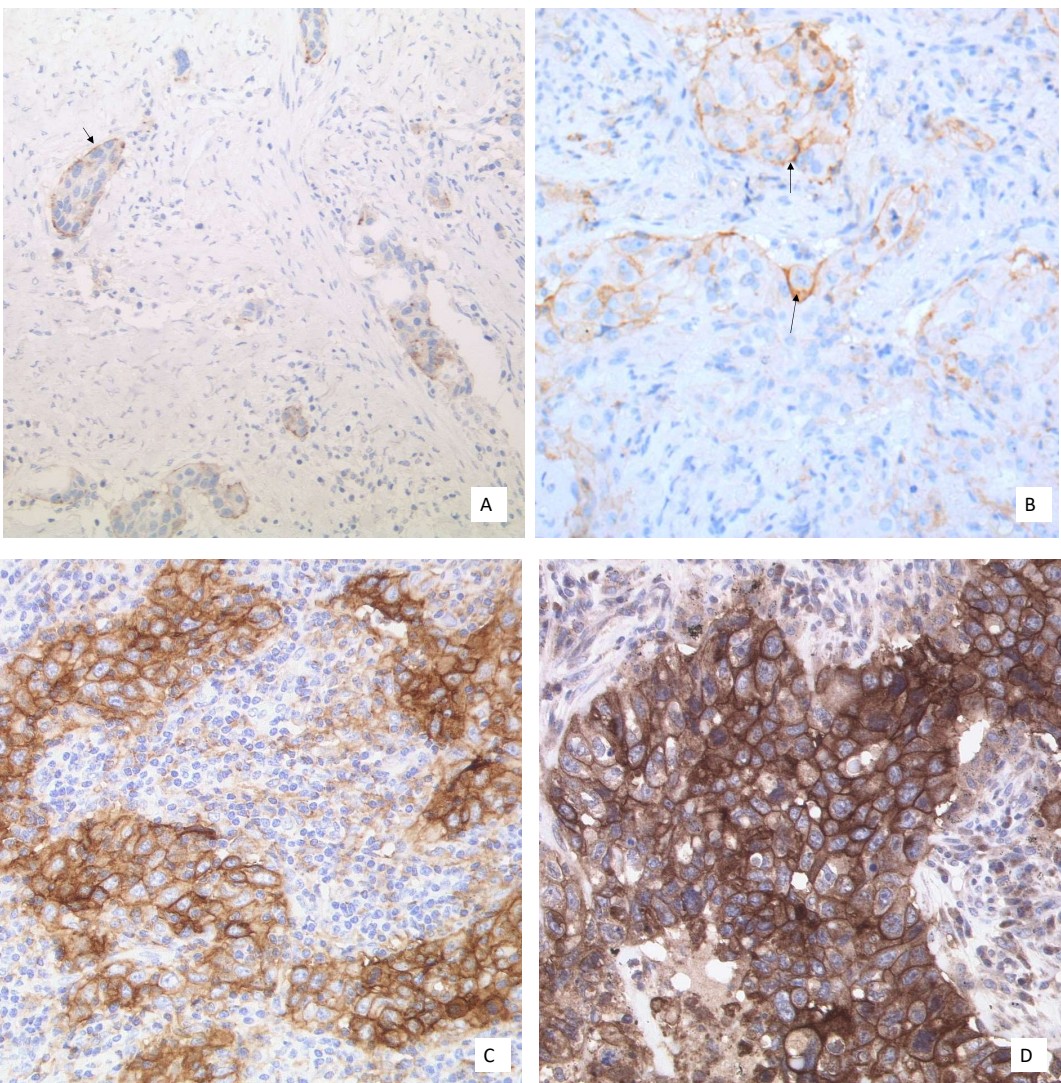

**Figure 1.** Different PD-L1 expression patterns in NSCLC tumor cells: (**A**) Basement membrane staining of moderate intensity (20× Obj. magnification); (**B**) Basolateral membrane staining (arrow) of weak to moderate intensity (40× Obj. magnification); (**C**) Moderate to strong membranous PD-L1 staining (40× Obj.); (**D**) Strong membranous PD-L1 staining (40× Obj.). (**B**,**C**) to be considered for scoring.

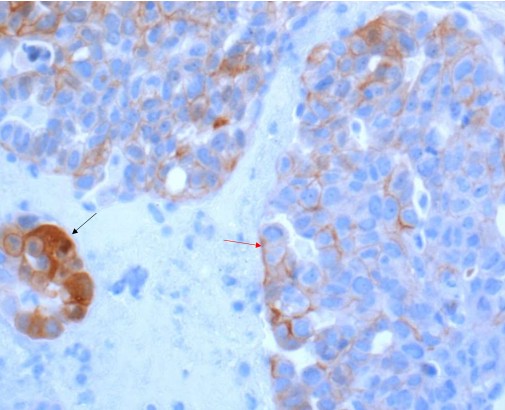

**Figure 2.** NSCLC with a heterogeneous PD-L1 expression pattern (40× Obj.) In some tumor cell areas, the membranous staining is incomplete (red arrow). A tumor cell nest with cytoplasmic staining to be excluded from scoring (black arrow).

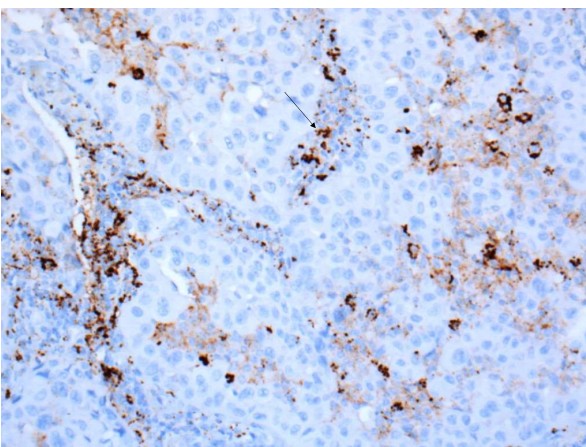

**Figure 3.** Tumor with PD-L1-positive immune cells, which often show a cytoplasmic staining (arrow) (40× Obj.).

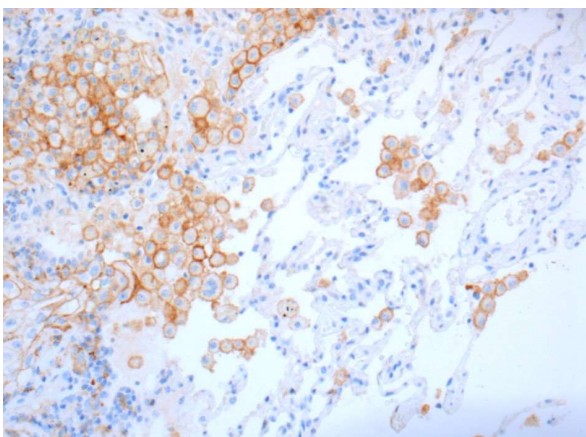

**Figure 4.** Positive stainings of alveolar macrophages are not counted but are useful as positive control (20× Obj.).

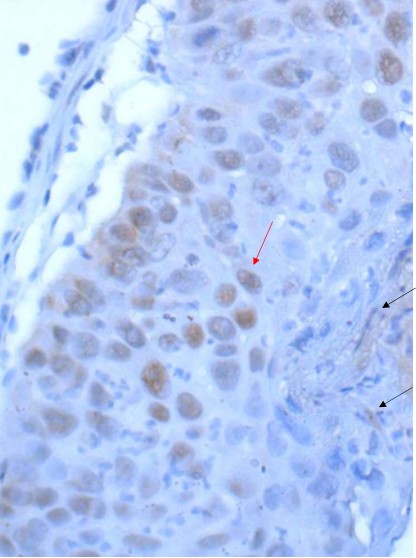

**Figure 5.** Squamous cell carcinoma of the lung with nuclear PD-L1 staining (red arrow), which is excluded from scoring as well as a weakly positive staining in stroma elements (black arrows) (40× Obj.).

The lower magnification is initially helpful for recognizing the distribution pattern of PD-L1 within the tumor and, if necessary, the heterogeneity of the staining. Nevertheless, the higher magnification is important for detecting the weakly stained tumor cells in order to be able to include them in the scoring.

The expression of PD-L1 can be heterogeneous in a carcinoma, so biopsies may well show different expression patterns. As a result, biopsies can show different expression patterns over time even without therapy. By the same argument, the lack of expression in a sample biopsy does not necessarily mean that the carcinoma is actually PD-L1-negative.

If there is a heterogeneous PD-L1 expression pattern, it can be helpful to subdivide the tumor tissue on the slide into several segments, determine the percentage tumor positivity in each of the segments, and then calculate the mean PD-L1 TPS or TC status. It is important that the tumor areas without positive membrane staining are also included as part of the denominator in the calculation and the different percentages are set in relation to the evaluated areas.

In case the cut-off is 1%, the reference of the Dako interpretation manual is especially helpful. As it is noted that when using the $20\times$ magnification and assuming a tumor cell diameter of 20μm for tightly clustered tumor cells in a tumor nest without significant stroma, 2500 cells are expected to be present in one $\times 20$ field of view fully occupied by tumor cells. Accordingly, at least 25 carcinoma cells must show complete or incomplete membranous staining in such a field to reach the cut-off of 1%.

In some cases, it can be difficult to distinguish the carcinoma cells from macrophages, whereby, as already mentioned, the alveolar macrophages must not be counted. An immunohistological staining against macrophages or a double staining with PD-L1 and an antibody against macrophages may be required to solve this issue.

All tumor areas included in the slide must be counted and the consideration of hot spots alone is not acceptable.

At least 100 vital carcinoma cells should be available for evaluation. If this is not the case, another block with more carcinoma cells should be re-stained if necessary, or it should be mentioned in the report and critically evaluated.

Evaluations have shown that about one third of NSCLC carcinoma cells show membranous PD-L1 expression either in more than 50%, in less than 1% or in more than 1% but less than 50% of the cells [13].

### 4.2. Training

The authors of the different international harmonization and concordance studies as well as meta-analyses find that external quality assessment and trainings are necessary to guarantee a national and international consistency and quality in the scoring and interpretation of PD-L1 expression patterns by proficiency tests and quality networks [14,15]. According to our own experience, it seems that after the specific reader training of TPS scoring in NSCLC, pathologists show a high level of concordance with some variation between different cut-offs. Accordingly, the overall interobserver agreement was 95.6% for TPS $\geq$ 1% and 87.3% for TPS $\geq$ 50% among $n$ = 751 pathologists, as observed in 85 globally performed training sessions [16,17].

### 4.3. Reporting the Results

The histological report should mention whether the tissue to be examined is suitable, i.e., whether more than 100 vital carcinoma cells were available for the immunohistological detection of PD-L1. Furthermore, in countries where the EMA is responsible for drug approval, the platform and antibody used should be stated [12].

Since pathologists mostly do not know which drug is given to the patients, it is useful to mention all scores that are approved for the corresponding entity. The calculated value or the raw score should also be noted in the report, instead of only the score achieved around the cut-off, in order to meet later possible approvals of future drug therapy. Optionally, all previously known cut-offs can also be listed, irrespective of the current approval situation.

If the pathologist wishes to evaluate the result of the PD-L1 expression in the tumor, it is important to include the reference to the corresponding drugs and, if applicable, to the line of therapy, in the case of multiple approvals.

## 5. Laboratory Specific aspects

### 5.1. Laboratory-Developed Tests versus Clinical Trial Tests

As pathology laboratories often use only one technical platform to perform a wide variety of IHC assays, LDTs are needed to make PD-L1 testing widely available. Since there are currently not only many different PD-L1 antibodies available for immunohistochemical studies but different platforms and varying concentrations, incubation times and temperatures being used in individual laboratories, there are numerous LDTs that ultimately may be used in the application of immunotherapeutics. Therefore, it is important that these different LDTs provide comparable results to the approved study KITs based on clinical trial assays (CTAs).

Different harmonization and concordance studies as well as meta-analyses have been performed. Scheel et al. [12,18] investigated four CTAs (IHC 22C3 pharmDx, IHC 28-8pharmDx, SP142 Ventana assay, SP263 Ventana assay) on a defined collective of cases comprised of lung adenocarcinomas and squamous cell carcinomas and found that 28-8 and 22C3 showed similar staining patterns in the carcinoma cells, whereas SP142 had a weaker and SP263 a more intensive staining. Both SP142 and SP263 stained immune cells more intensely than 28-8 and 22C3. Similar results were obtained by Hirsch et al. [19], where the SP263 Ventana assay showed a similar staining to 28-8 pharmDx and 22C3 pharmDx, and only the SP142 Ventana assay was concluded as having a weaker staining in tumor cells.

In the Blueprint Phase 2 project [20], the study assay 73-10, which was added to the other assays, showed an even greater sensitivity over the other assays.

In the meta-analysis by Büttner et al. [21], it is stated that 28-8, 22C3 and SP263 showed a high concordance in tumor cells but not for the assessment of the PD-L1 expression in immune cells. Nevertheless, the LDTs required a standardization before they could be used in routine diagnostics. The use of reference samples with known PD-L1 expression levels, including cell line microarrays with defined levels of PD-L1 expression, helped to confirm the sensitivity and specificity of each of the LDTs [14].

External quality assessment (EQA) programs have been offering proficiency tests specifically for PD-L1 immunostaining for several years. Accordingly, the German EQA scheme (QuIP) was able to show that with accurate validation, good staining results comparable to the assays used in the clinical approval trials (CTAs) could be achieved by the LDTs.

### 5.2. Controls

The use of adequate staining controls is of paramount importance. Either commercially available cell lines or well-fixed tonsils (Figure 6A) should be selected as positive controls for PD-L1 staining. The advantage of tonsils as a positive control is the presence of both immune and epithelial cell staining. Whereas most of APCs (>50%) in the activated germinal centers should show at least a weak staining (Figure 6B), epithelial cells show strong membrane staining only in the deep crypts where lymphoid cells enter the "reticulated" epithelium (Figure 6C) [22]. This described immunohistochemical pattern does not apply to the Ventana Assay SP142. In this assay, the difference in staining intensity does not play a significant role. PD-L1 staining is often dark brown with a punctate or linear staining pattern. Regardless of the different LTDs, the surface epithelium, the endothelium as well as the fibroblasts should not be stained (manual).

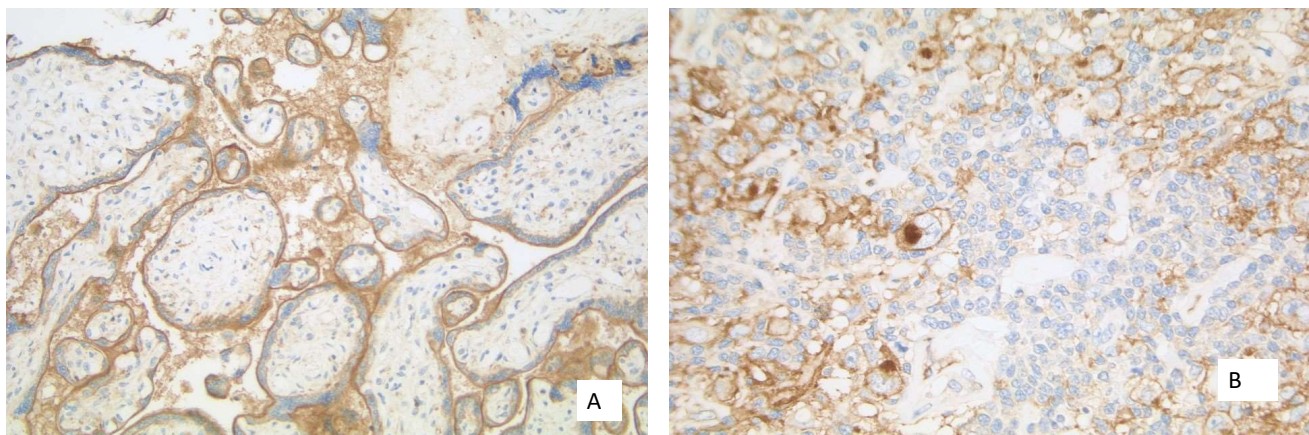

**Figure 6.** (**A**) Well-fixed tonsil exhibiting PD-L1 expression at two different sites (20× Obj.); (**B**) Moderate expression in the APCs of the germinal centers (40× Obj.); (**C**) Strong expression in the epithelium of the deep crypts (40× Obj.).

Placental tissue (Figure 7A) and the tissue of classical Hodgkin lymphomas (Figure 7B) should not be used for establishing controls because these tissues do not have different staining intensities.

**Figure 7.** (**A**) Placental tissue with membranous staining of the syncytiotrophoblast (40× Obj.); (**B**) Hodgkin cells with expression of PD-L1 at the membrane and Golgi region (40× Obj.).

The use of a negative control is not only important from a quality point of view, but also very helpful, especially in the case of NSCLC, as it allows the anthracotic areas to be identified, which facilitates the evaluation of the sample.

If cell pellets are used as a positive control, it must be taken into account here that the positive staining also refers to the quantity of stained cells and not only to the staining intensity. According to the definition, this means that individual cells in the negative control may be weakly positive, provided that the cut-off of $\geq 1$ intensity is not exceeded.

*5.3. Pre-Analytic*

An important factor influencing the immunohistological staining concerns the pre-analytics regarding formalin-embedded tissue. This process includes not only the cold ischemia time, but also the fixation medium, the fixation duration, the thickness of the sections as well as the choice of slides.

In the context of pre-analytics in tissue processing, cold ischemia is the time between the removal of the tissue and the start of the formalin fixation [23]. This time lag has a significant effect on the quality of the tissue, as a shorter ischemia period will positively impact the preservation status of the tissue. Another important factor that affects further immunohistological examinations concerns the formalin fixation. Here, not only the quality of the formalin is important, but also the correct ratio between tissue volume and fixation quantity as well as the duration of fixation and the fixation temperature. There are numerous studies that have addressed this issue. Buesa et al. [24] studied this parameter in a collective of 60 tissue samples from the uterus, breast, liver, abdominal skin and underlying pure fat with a tissue size of 1 cm $\times$ 1 cm $\times$ 3 mm thickness. A ratio of 2:1 for the volume of neutral buffered formalin (NBF) to 3 mm thick tissue samples seemed adequate to assure a 50% cross-linking after 48 h at an ambient temperature of 25 °C, which is present in most laboratories.

Goldstein et al. [23] studied different samples of estrogen receptor-positive invasive breast carcinomas, where the sample size was considered. Thus, they selected disparate and similar needle core biopsies, as well as partial mastectomy specimens. They found that regardless of the size of the tissue samples examined, the samples must be fixed for 6–8 h to obtain reliable immunohistological results.

Finally, the choice of slides for the immunohistochemical staining also plays an important role. For this examination technique, slides with special adhesive substances should be used to prevent tissue detachment.

## 6. Tissue-Specific Aspects and Clinical Course

*6.1. Tissue Samples versus Cytology*

All approval studies were performed on tissue samples. Nevertheless, as the diagnosis of lung cancer in the advanced stage is often made on cytological specimens, the question whether cytological samples are also useful for PD-L1 determination was discussed [25,26]. Bubendorf et al. [27] investigated the PD-L1 expression in cytology and histology specimens from patients with NSCLC. They collected paired cytology and tissue samples from 190 patients and stained them with the antibodies SP142 and SP263 using Benchmark ultra to detect PD-L1. The result of this study was that a qualitative PD-L1 analysis was feasible for most of the cytology samples when a cell block was prepared. However, the quantitative analysis of PD-L1 using cytological material was challenging, especially because of the low tumor and immune cell content and the lack of tissue context [28]. For this reason, tissue samples are preferred for the investigation of PD-L1 status. Only if no tissue samples but cytology samples are available for PD-L1 testing may cytological preparations be utilized for the immunohistological analysis of PD-L1, assuming either small tissue fragments are available or paraffin blocks could be prepared from them [29]. Smear preparations are not suitable for determining PD-L1 expression [22].

*6.2. Primary Tumor versus Metastasis*

Studies have demonstrated that in synchronous tumor manifestations, the heterogeneity of PD-L1 expression is pronounced enough to be clinically relevant in only 10% of cases studied. Munari et al. [30] investigated, in this immunohistological study, the PD-L1 expression in 84 cases of paired primary and relapsed tumor samples from patients with NSCLC using 1% and 50% cut-offs. The authors found a PD-L1 expression concordance between primary and paired metastatic tumor in 88% and 90%. In this respect, the choice of material between the primary tumor and metastasis was not a crucial issue.

*6.3. Clinical Course*

A more important factor, however, is the PD-L1 expression considered over the time course of the patient, as both radio- and chemotherapy have an influence on the PD-L1 expression [31]. Therefore, PD-L1 testing should be performed, if possible, on the most recent material and re-biopsy should be considered if necessary [22,30]. Moreover, Munari et al. [30] described, in the same study mentioned above, a lower concordance (66%) regarding local tumor recurrences.

**7. Further Developments**

In spite of the present development, not all patients with a positive PD-L1 status could benefit from treatment with immunotherapy. Therefore, further efforts are being made to optimize the therapy. For example, it has been demonstrated that the tumor mutation burden (TMB) in concert with the PD-L1 expression seemed to be a useful biomarker for immune checkpoint blockade [7].

Furthermore, Kong et al. [32] showed that the cell-surface adhesion receptor CD44 is a key positive regulator of the PD-L1 expression in NSCLC and triple negative breast cancer (TNBC). The positive detection of CD44 correlated with the PD-L1 expression at the mRNA and protein level in primary tumor samples.

Several studies correlating the PD-L1 expression with driver mutation have revealed different results. Some cell line studies found a positive correlation between PD-L1 expression and EGFR mutations, while other authors demonstrated an association between PD-L1 expression and EGFR wildtype. Other studies did not find any correlation between PD-L1 and EGFR mutations [31,33,34].

Peng et al. [35] investigated the EGFR-TKI resistance and demonstrated that this resistance promoted immune escape in lung cancer by increasing the PD-L1 expression. The analysis based on TCGA datasheets and paired NSCLC patients before and after EGFR-TKI resistance described HGF, MET-amplification and EGFR-T790M to upregulate the PD-L1 expression.

Gao et al. [36] correlated the IFN-gamma-mediated inhibition of lung cancer to PD-L1 expression and PI3K-AKT pathway activation. The author and his team observed that the activation of the JAK2-STAT 1 pathway seemed to be responsible for the antiproliferative effect of IFN-gamma. The inhibition of PI3K led to the downregulation of PD-L1 expression and enhanced the antiproliferative effect of IFN-gamma.

Hurkmans et al. [37] investigated the tumor mutational load (TML), CD8+ T-cells, expression of PD-L1 and HLA class I. High TML, high CD8+ T-cells and no loss of HLA class I as well as high PD-L1 expression were associated with a better PFS and suggested that these were better predictive biomarkers for a response to anti-PD-1 immunotherapy.

To summarize, immunotherapy has changed the treatment of NSCLC fundamentally, making the immunohistological detection of PD-L1 an often-used basis for decisions concerning immunotherapy. PD-L1 is therefore of great importance as a predictive biomarker. The establishment of the antibody requires careful validation as well as the selection of good controls, good pre-analytics, and finally, internal and external quality assessment. The interpretation of the staining also requires training. Since not all patients benefit from immunotherapy, further research approaches are needed to validate new predictive markers and optimize existing ones.

**Author Contributions:** Conceptualization, K.J. and J.R.; writing—original draft preparation, K.J.; writing—review and editing, K.J. and J.R.; All authors have read and agreed to the published version of the manuscript.

**Funding:** This research received no external funding.

**Acknowledgments:** Our special thanks go to Bharat Jasani, Director of Pathology, Targos Molecular Pathology GmbH, Kassel, for revising the manuscript as well as to Lina Yap project manager from QuIP for english editing.

**Conflicts of Interest:** The authors declare no conflict of interest.

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
