# Peer review of "The Challenge to the Pathologist of PD-L1 Expression in Tumor Cells of Non-Small-Cell Lung Cancer—An Overview"

_curroncol, doi:10.3390/curroncol28060437_

Round 1

Reviewer 1 Report

It seems that the title and the content of the main text do not match.

There are few citations for review. There are no references in the sections of "General aspects in interpretation of PD-L1 staining", "Reporting", and "Controls". For this reason, it is not described in detail how the described procedure was established.

It is very difficult to read because there are few figures and tables.

In the section of “General aspects in interpretation of PD-L1 staining”, please show examples such as “Basolateral pattern of membrane staining”, “cytoplasmic staining”, “nuclear expression”, “staining of stroma elements” or “basement membrane”.

Author Response

Thank you for the constructive comments, which we have implemented as follows:

  1. There are few citations for review. There are no references in the sections of "General aspects in interpretation of PD-L1 staining", "Reporting", and "Controls". For this reason, it is not described in detail how the described procedure was established.

We add more references especially in the mentioned sections

  1. It is very difficult to read because there are few figures and tables.

We´ve added 7 figures and modified the table.

  1. In the section of “General aspects in interpretation of PD-L1 staining”, please show examples such as “Basolateral pattern of membrane staining”, “cytoplasmic staining”, “nuclear expression”, “staining of stroma elements” or “basement membrane”.

Figure 1 demonstrate the basolateral and basement membrane pattern, figure 3 the cytoplasmic staining, figure 5 the nuclear staining and stroma elements. Moreover we´ve added figures with PD-L1 expression at alveolar macrophages as well as positive controls.

please see the attachments

Reviewer 2 Report

This review gave an overview on the roles of PD-L1 expression in tumor cells of Non-small cell lung cancer. Both the writing style and the content need minor changes for this article to be useful to readers of the journal.

 Comments:

  1.        The full name of NSCLC, PD1, PD-L1 and TPS in abstract section should be added.
  2.         Review articles typically have a high-quality Figure illustration to provide a broad overview of the topic. This review has no figures. Perhaps the table should be converted into plots.
  3.         The order and format of references used need to check.

Author Response

Thank you for the constructive comments, which we have implemented as follows:

  1. The full name of NSCLC, PD1, PD-L1 and TPS in abstract section should be added.

We add the full names of NSCLC, PD1, PD-L1 and TPS in the abstract.

  1. Review articles typically have a high-quality Figure illustration to provide a broad overview of the topic. This review has no figures. Perhaps the table should be converted into plots.

We included 7 pictures of different expression patterns in NSCLC and control tissue, additionally we modified the table accordingly.

  1. The order and format of references used need to check.

We checked the order and format of the references

Round 2

Reviewer 2 Report

I recommend to publish in this present form